# Genome-Wide Identification of the Ferric Chelate Reductase (*FRO*) Gene Family in Peanut and Its Diploid Progenitors: Structure, Evolution, and Expression Profiles

**DOI:** 10.3390/plants13030418

**Published:** 2024-01-31

**Authors:** Junhua Guan, Zheng Zhang, Gangrong Shi

**Affiliations:** College of Life Sciences, Huaibei Normal University, Huaibei 235000, China

**Keywords:** *Arachis hypogaea*, *FRO*, cultivar, Fe deficiency, Cu accumulation

## Abstract

The ferric chelate reductase (*FRO*) family plays a vital role in metal ion homeostasis in a variety of locations in the plants. However, little is known about this family in peanut (*Arachis hypogaea*). This study aimed to identify *FRO* genes from the genomes of peanut and the two diploid progenitors (*A. duranensis* and *A. ipaensis*) and to analyze their gene/protein structures and evolution. In addition, transcriptional responses of *AhFRO* genes to Fe deficiency and/or Cu exposure were investigated in two peanut cultivars with different Fe deficiency tolerance (Silihong and Fenghua 1). A total of nine, four, and three *FRO* genes were identified in peanut, *A. duranensis*, and *A. ipaensis*, respectively, which were divided into three groups. Most *AhFRO* genes underwent WGD/segmental duplication, leading to the expansion of the *AhFRO* gene family. In general, clustered members share similar gene/protein structures. However, significant divergences occurred in *AhFRO2* genes. Three out of five *AhFRO2* genes were lowly expressed in all tissues under normal conditions, which may be beneficial for avoiding gene loss. Transcription analysis revealed that *AhFRO2* and *AhFRO7* genes might be involved in the reduction of Fe/Cu in plasma membranes and plastids, respectively. *AhFRO8* genes appear to confer Fe reduction in the mitochondria. Moreover, Fe deficiency induced an increase of Cu accumulation in peanut plants in which *AhFRO2.2/2.4/2.5* and *FRO7.1/7.2* might be involved. Our findings provided new clues for further understanding the roles of *AhFRO* genes in the Fe/Cu interaction in peanut.

## 1. Introduction

Iron (Fe) is a microelement that is essential for plant growth and development. In plants, Fe functions as a constituent of many important molecules such as Fe-sulfur (Fe-S) and heme Fe proteins, which are involved in fundamentally biological processes, including respiration, photosynthesis, chlorophyll biosynthesis, sulfur assimilation, and nitrogen fixation [1]. The function of Fe is mainly based on the reversible redox reaction of ferrous (Fe^2+^) and ferric (Fe^3+^) ions and its ability to form octahedral complexes with various ligands. Fe deficiency not only inhibits chlorophyll synthesis and reduces photosynthetic efficiency [2] but also interrupts the respiratory electron transport and tricarboxylic acid cycle [3]. Meanwhile, Fe in excess can be toxic because free Fe ions induce the formation of reactive oxygen species via the Fenton reaction [4,5]. Therefore, cellular Fe levels must be strictly controlled in plants.

Although Fe is the fourth most abundant element in the earth’s crust, it is not easily taken up by plants due to the predominance of insoluble ferric oxides in soils, particularly in calcareous soils that account for approximately 30% of the world’s arable soils [6]. Consequently, crops grown in calcareous soils often suffer from iron deficiency, limiting yield and quality. Moreover, Fe deficiency in plants also poses serious health problems because plant foods are the main source of dietary Fe for humans. It is estimated that 30%–50% of anemia in children and other groups is caused by iron deficiency [7]. Thus, understanding plant iron homeostasis is essential for improving crop yield and human iron nutrition.

Plants have evolved complex mechanisms to sense and respond to iron fluctuations in the rhizosphere and prevent iron deficiency or toxicity by maintaining Fe homeostasis [4,5,8]. Non-gramineous plants use the reduction strategy (strategy I) for Fe acquisition, while gramineous plants adopt the chelation strategy (strategy II). The reduction strategy includes three processes: (i) releasing protons from root cells to the rhizosphere via H^+^-ATPase for reducing soil pH value; (ii) reducing Fe^3+^ to Fe^2+^ by ferric chelate reductases (FRO) in the acidifying rhizosphere; and (iii) taking up Fe^2+^ into root cells through iron-regulated transporter 1 (IRT1) in the plasma membrane. Gramineous plants can secrete the mugeneic acids (MAs) like phytosiderophores from their roots to the soil to dissolve Fe^3+^ in the rhizosphere to form a Fe^3+^–MA complex and then absorb the Fe^3+^-MA complex into root cells through yellow stripe like protein (YSL) in the plasma membrane.

The FRO family plays a vital role in metal ion homeostasis in a variety of locations in plants [9]. It belongs to the flavocytochrome superfamily that transfers electrons across membranes [10]. FRO proteins contain eight transmembrane helices and share three typical domains, a heme-containing ferric reductase domain (PF01794) in the transmembrane region, and the FAD-binding (PF08022) and NAD-binding (PF08030) domains inside the membrane of the C-terminal region [11,12]. In *Arabidopsis*, AtFRO2 is responsible for the reduction of solubilized Fe^3+^ to Fe^2+^ at the root surface, where Fe^2+^ is then transported into the cytoplasm via AtIRT1 in the root plasma membrane [10,13]. AtFRO6 mediates the reduction of Fe^3+^ to Fe^2+^ at the plasma membrane of leaf cells [9,14]. AtFRO7 plays a role in chloroplast iron acquisition by reducing Fe^3+^ to Fe^2+^ [15,16]. AtFRO3 and AtFRO8 have been predicted to localize to mitochondrial membranes and might serve an analogous function in the mitochondrial iron homeostasis [16,17]. While several *FRO* genes were functionally characterized in *Arabidopsis*, little is known about the roles of this family in other plant species including peanut (*Arachis hypogaea* L.), a major oil-seed crop mainly grown in temperate and tropical regions of the world.

In this study, based on the whole-genome sequences [18,19], *FRO* family genes of the peanut (cv. Tifrunner) and the two wild ancestral species (*A. duranensis* and *A. ipaënsis*) were identified, and the structures, functions and evolutionary relationships were characterized. Moreover, the expression of *AhFRO* genes in response to Fe deficiency and/or Cu exposure was investigated. Our data would provide a basis for further functional characterization of *AhFRO*s and shed new light on the possible roles of the *AhFRO* family in the uptake and translocation of Fe and Cu in plants.

## 2. Results

### 2.1. Identification and Phylogenetic Analysis of FRO Genes in the Three Arachis Species

Analysis of BLASTp using AtFROs as queries resulted in 26, 14, and 11 non-redundant protein sequences from genomes of *A. hypogaea* cv. Tifrunner, *A. duranensis*, and *A. ipaënsis*, respectively. Phylogenetic analysis indicated that these proteins were divided into two clades: one including all eight AtFRO members could be considered as the FRO family, and the other might be respiratory burst oxidases (Appendix A). A total of nine putative *AhFRO* genes were identified in peanut, including five *AhFRO2* (*AhFRO2.1/2.2/2.3/2.4/2.5*), two *AhFRO7* (*AhFRO7.1/7.2*), and two *AhFRO8* (*AhFRO8.1/8.2*, Table 1). Meanwhile, five *AdFRO* (*AdFRO2.1/2.2/2.3*, *AdFRO7*, and *AdFRO8*) and three *AiFRO* (*AiFRO2*, *AdFRO7*, and *AdFRO8*) genes were identified from *A. duranensis* and *A. ipaënsis*, respectively (Appendix A and Table 1).

The length of *FRO* genes varied from 1254 bp (*AhFRO2.4*) to 2999 bp (*AhFRO7.1*), from 556 bp (*AdFRO2.3*) to 2508 bp (*AdFRO7*), and from 2446 bp (*AiFRO8*) to 2887 bp (*AiFRO7*) for *A. hypogaea*, *A. duranensis*, and *A. ipaënsis*, respectively. The CDS length varied from 702 bp (*AhFRO2.4*) to 2217 bp (*AhFRO7.2*), from 372 bp (*AdFRO2.3*) to 2217 bp (*AdFRO7*), and from 2124 bp (*AiFRO8*) to 2220 bp (*AiFRO7*) for *A. hypogaea*, *A. duranensis*, and *A. ipaënsis*, respectively. The number of amino acids varied from 233 (AhFRO2.4) to 738 (AhFRO 7.2), from 123 (AdFRO2.3) to 738 (AdFRO7), and from 707 (AiFRO8) to 739 (AiFRO7) for AhFRO, AdFRO, and AiFRO proteins, respectively. The molecular weight varied from 25.97 kDa (AhFRO2.4) to 83.53 kDa (AhFRO2.2), from 13.52 kDa (AdFRO2.3) to 83.14 kDa (AdFRO7), and from 79.36 kDa (AiFRO8) to 83.21 kDa (AiFRO7) for AhFRO, AdFRO, and AiFRO proteins, respectively. The instability index for most FRO proteins was larger than 40, suggesting a low stability in vitro. All FRO proteins showed a high aliphatic index (91.87–112.43), implying a high stability over a wide temperature range. The GRAVY of all FRO proteins except AdFRO2.3 is higher than 0, indicating that FROs are hydrophobic proteins. Most of the FRO proteins are basic proteins (pI > 7) (Table 1). The number of TMDs for most FRO ranged from 8 to 12, while AhFRO2.4 and AdFRO2.2 had two TMDs. No TMD was detected in AdFRO2.3 (Table 1). FRO2 proteins were predicted to be localized in plasma membranes, while FRO7 and FRO8 were localized in chloroplast and mitochondria, respectively (Table 1).

To comprehensively dissect the evolutionary relationship of the *FRO* gene family, a phylogenetic analysis was carried out on 35 FRO proteins from *A. hypogaea*, *A. duranensis*, *A. ipaënsis,* and other four plant species (Figure 1). As shown in Figure 1, the FRO proteins could be classified into three groups. Group I, which is signed by five *Arabidopsis* AtFROs (AtFRO1–5), includes five AhFROs (AhFRO2.1/2.2/2.3/2.4/2.5), three AdFROs (AdFRO2.1/2.2/2.3), and one AiFROs (AiFRO2) from peanut, *A. duranensis*, and *A. ipaënsis*, respectively. Group II is signed by two *Arabidopsis* AtFROs (AtFRO6/7) and is composed of AhFRO7.1/7.2, AiFRO7, and AdFRO7. Group III consists of AtFRO8, AhFRO8.1/8.2, AiFRO8, and AdFRO8. By contrast, the three *Arachis* species showed closer relationships with another legume species (*M. truncatula*) in terms of FRO proteins.

### 2.2. Conserved Motifs, Domain Architectures, and Exon–Intron Organization

A total of ten conserved motifs (1–10) were identified in FRO proteins, with the length varying from 21 to 50 (Figure 2A and Appendix A). The majority of FRO proteins contained the ten conserved motifs. However, AhFRO2.4, AdFRO2.2, and AdFRO2.3 only contained four, three, and two motifs, respectively. The composition of conserved motifs was similar within phylogenetic groups. All FRO proteins contained the typical domains (Ferric_reduct, FAD_binding_8, and NAD_binding_6) except AhFRO2.4, AdFRO2.2, and AdFRO2.3 in which only the NAD_binding_6 domain was detected (Figure 2B).

Multiple sequence alignment indicated that all AhFROs have conserved motifs such as C(L/M)AxL, YHxWLG, and HG in the Ferric_reduct domain, LQWH(P/S)F in the FAD_binding domain, and GGxG(I/L)(T/S)PF in the NAD_binding domain (Appendix A). In addition, some conserved motifs including LxxGL, FExFxYxHxLY, LRxxQS, VxIK, EGPY(G/E), and GV(L/F)(V/A)(C/S)GP were also detected in other regions.

To gain insight into the evolution of the *FRO* family in peanut, exon–intron organizations were examined (Figure 2C). *FRO* genes typically contained eight or nine introns, which were separated by seven or eight exons, while only two exons were detected in *AhFRO2.4*, *AdFRO2.2*, and *AdFRO2.3*. The exon–intron organization varied among different phylogenetic groups; however, *FRO* genes belonging to the same group generally had similar structures. 

### 2.3. Gene Duplication of the FRO Family

The nine *AhFRO* genes of peanut were distributed in six chromosomes (Ah02, 04, 07, 12, 14, and 17). The sub-genomes A (Ah01-10) and B (Ah11-20) have four and five *AhFRO* genes, respectively (Figure 3A). The number of *AhFRO* genes was the highest in Ah14 (3), followed by Ah04 (2), while only one gene was contained in Ah02, 07, 12, and 17, respectively. Similarly, five *AdFRO* genes of *A. duranensis* were distributed in A02, A04, and A07, while *AiFRO* genes of *A. ipaënsis* were distributed in B02, B04, and B07 (Figure 3B,C).

Synteny analysis revealed that the four *AhFRO* genes of the sub-genome A were crossly collinear with corresponding genes of the sub-genome B, forming four orthologous gene pairs derived from whole-genome duplications (WGDs): *AhFRO2.1/2.3*, *AhFRO2.2/2.4*, *AhFRO7.1/7.2*, and *AhFRO8.1/8.2* (Figure 3A). In addition, segmental duplication also occurred within the two sub-genomes, forming two paralogous gene pairs (*AhFRO2.1/2.2* and *AhFRO2.3/2.4*) (Figure 3A).

To better understand the evolution of the *FRO* gene family, an interspecies synteny analysis was performed on the three *Arachis species*. Three orthologous gene pairs (*AdFRO2.1/AiFRO2*, *AdFRO7/AiFRO7,* and *AdFRO8/AiFRO8*) were identified between *A. duranensis* and *A. ipaënsis*, which is less than that between the two sub-genomes of peanut (Figure 3B). In addition, a block (*AdFRO2.1/AdFRO2.3*) within the genome of *A. duranensis* appears to be WGD/segmental duplication. There were ten and eight collinear blocks between peanut and *A. duranensis* and between peanut and *A. ipaënsis*, respectively (Figure 3C). All *FRO*s of *A. duranensis* and *A. ipaënsis* were collinear with those of peanut.

The *K*a/*K*s ratios of all gene duplication pairs were greatly lower than one (Table 2), indicating that *AhFRO* genes evolved under purifying selection [20]. The divergence time of the four whole genome duplicated gene pairs ranged from 1.21 Mya to 2.38 Mya, which was considerably less than that of the two segmental duplicated gene pairs (43.14 and 44.70 Mya, respectively) (Table 2).

### 2.4. 3D Model Predictions and Multiple Sequence Alignment

To obtain a reasonable theoretical structure of FROs, 3D model predictions were performed using the Swiss-Model server (Figure 4 and Appendix A). Most of the FRO2 proteins in peanut and the progenitors were well modeled with the homologous template, 7d3f.1, which is a cryo-EM structure of human DUOX1–DUOXA1 in a high-calcium state (Figure 4 and Appendix A). All FRO7 were well modeled with 6wxr.1, the cryoEM structure of mouse DUOX1–DUOXA1 complex in the absence of NADPH, while FRO8 was well modeled with 8gz3.1, the structure of human phagocyte NADPH oxidase in the resting state. Apart from three short sequence proteins (AhFRO2.4, AdFRO2.2, and AdFRO2.3), all FRO proteins from peanut and the progenitors share more than 20% sequence identity with their homologous templates, and the GMQE values ranged from 0.29 to 0.43 (Appendix A), suggesting a high reliability of 3D model predictions.

### 2.5. The Cis-Regulatory Elements (CREs) of AhFRO Genes in Peanut

A total of 1040 CREs were identified in the promoter region of *AhFRO* genes, and most of them are associated with gene transcription, light response, phytohormone response, and abiotic stress (Table 3). The main light-responsive CREs are TCT-motif, Box 4, ATCT-motif, GT1-motif, G-box, and AT1-motif. The main phytohormone-responsive CREs included ABRE, P-box, and TCA-element. The abiotic stress-responsive CREs are TC-rich repeats, ARE, LTR, and MBS. The promoter of all *AhFRO* genes contained CAAT-box, TATA-box, and TCT-motif; however, the distribution of remaining CREs widely varied among *AhFRO* genes. *AhFRO2.3* contained the most light-responsive CREs, while *AhFRO8.1* had the most phytohormone-responsive elements. *AhFRO2.2* and *AhFRO2.5* have the fewest types of CREs (Table 3).

### 2.6. Tissue-Specific Expression of AhFRO Genes in Peanut

To gain an insight into tissue-specific expression, RNA-seq data of the nine *AhFROs* were used for studying their expression patterns in different tissues and developmental stages (Appendix A). As presented in Figure 5, nine *AhFRO* genes were divided into three clusters. Cluster I included *AhFRO7.1* and *AhFRO7.2,* which show high expression and are mainly transcribed in leaves and pistils. Cluster II is composed of four genes with an intermediate level of expression that is preferentially expressed in the developing seeds, roots, vegetative shoot tip, and mainstem leaves. Cluster III consists of *AhFRO2.1*, *AhFRO2.2*, and *AhFRO2.4,* which show low expression and were predominantly expressed in roots.

### 2.7. Transcriptional Responses of AhFROs to Fe-Deficiency and Cu Exposure

To elucidate the transcriptional response of *AhFRO*s to Fe deficiency and/or Cu exposure, two contrasting peanut cultivars, Fenghua 1 (Fe deficiency sensitive cultivar) and Silihong (Fe deficiency tolerant cultivar), were used for qRT-PCR analysis. As presented in Figure 6, Cu exposure repressed the expression of *AhFRO7.1/7.2* in the root for both cultivars, while *AhFRO2* genes were not affected. Fe deficiency induced the expression of *AhFRO2.1/2.2/2.3/2.5* but reduced the expression of *AhFRO7.1* in the root for both cultivars. The remaining *AhFRO* genes responded Fe deficiency in a cultivar-specific manner. Cu exposure with Fe deficiency increased the expressions of *AhFRO2.1/2.2/2.4/2.5* but repressed the expression of *AhFRO7.1/7.2* in the root for both cultivars, while the expression of *AhFRO8.1/8.2* was unaffected (Figure 6). 

As for the gene expression in leaves, *AhFRO2.2*, *AhFRO7.1/7.2*, and *AhFRO8.1* were repressed by Cu exposure for both cultivars (Figure 7). Fe deficiency induced the expression of *AhFRO2.1/2.2/2.4/2.5* but reduced the expression of *AhFRO8.1/8.2* in the leaves for both cultivars. Cu exposure with Fe deficiency up-regulated the expressions of *AhFRO2.2/2.4/2.5* in the leaves for both cultivars (Figure 7).

### 2.8. The Accumulation and Translocation of Fe and Cu in the Two Peanut Cultivars

The two peanut cultivars differed from each other in Fe accumulation, which was significantly influenced by Fe deficiency and Cu exposure as well as their interactions (Table 4). Under normal conditions (control), Fenghua 1 showed higher Fe concentrations in roots and shoots, and higher total amounts of Fe in plants than Silihong (Table 4). Fe deficiency significantly reduced Fe uptake and accumulation in the peanut plant depending on cultivar and Cu exposure. Cu exposure significantly increased root Fe concentrations in Fe-sufficient peanut plants, resulting in an increase in total amounts of Fe in plants and a reduction of root-to-shoot Fe translocation (Table 4).

The two peanut cultivars are similar in Cu accumulation and translocation (Table 4). Cu concentrations in roots and shoots and total amounts of Cu in plants were significantly enhanced by Fe deficiency and Cu exposure, while the percentage of Cu in shoots was reduced (Table 4). There are significant Cu × Fe interactions on Cu accumulation and translocation in the two peanut cultivars (Table 4). 

### 2.9. Relationships between AhFRO Genes and Metal Accumulation in Peanut

Pearson’s correlation analysis was performed to determine relationships between *AhFRO* genes and the accumulation and translocation of Fe and Cu. As shown in Table 5, the expression of all *AhFRO2* genes was negatively correlated with Fe concentrations in roots (*p* < 0.05) and shoots (*p* < 0.01) as well as the total Fe in plants (*p* < 0.05). In contrast, *AhFRO7.1* and *AhFRO8.2* were observed to positively correlate with Fe accumulation (*p* < 0.05). Cu accumulation in peanut plants was positively correlated with the expression of *AhFRO2.2/2.4/2.5* (*p* < 0.01) but negatively correlated with the expression of *AhFRO7.1/7.2* (*p* < 0.05). The percentage of Cu in shoots was negatively related to the expression of *FRO2.2/2.5* (*p* < 0.01) but positively correlated with the expression of *AhFRO7.1/7.2* (*p* < 0.01). No significant correlation was found between *AhFRO* genes and Fe translocation (Table 5).

## 3. Discussion

FRO members have been demonstrated to play crucial roles in the homeostasis of Fe and Cu [9]. However, there has been little work on genome-wide identification of the *FRO* family in plants. In this study, we identified nine, four, and three *FRO* genes in peanut, *A. duranensis*, and *A. ipaensis*, respectively (Table 1). The number of *AhFRO* genes in peanut is higher than that in most reported plant species [9,12]. The same phenomenon has been reported in other gene families of peanut [22,23,24,25]. Peanut, as an allotetraploid species derived from the hybridization of diploid ancestral species, *A. duranensis* (AA) and *A. ipaensis* (BB) [19], has experienced at least three rounds of WGD events [26]. Our results indicated that eight out of nine *AhFRO* genes have experienced WGD events. Moreover, two paralogous gene pairs (*AhFRO2.1*/*2.2* and *AhFRO2.3*/*2.4*) were found to be segmental duplications. Expectedly, the divergence time indicates that segmental duplication events (43.14–44.70 Mya) of *AhFRO* genes occurred dramatically earlier than WGD (1.21–2.38 Mya) (Table 2). It is likely that WGD/segmental duplication contributes to the expansion of the *AhFRO* gene family in peanut.

Gene duplication is a major source of novel genes that contribute to the acquirement of novel functions [27]. However, it results in functional redundancy [28] and, consequently, most duplicated genes quickly pseudogenize and get lost [29]. In the current study, we found that the number of *FRO* genes differed between the two sub-genomes of peanut and between *A. duranensis* and *A. ipaensis*, which suggests an asymmetrical evolution in the family. Synteny analysis revealed that the orthologs of *AhFRO2.5* and *AhFRO2.3* have been lost in the genome of *A. ipaensis* after allopolyploidization (Figure 3). Likewise, an ortholog of *AdFRO2.3* has been lost in the sub-genome A of peanut. These results, which are in agreement with our previous study [25], confirmed that gene loss is easier in *A. ipaensis* than *A. duranensis*. The number of *AhFRO* genes in peanut is greater than the sum of the two ancestors, suggesting that heteropolyploid is more capable of avoiding gene loss than diploid.

Another approach for avoiding gene loss of duplicated genes is the reduction of their expression compared to the ancestral gene [28]. In the present study, three *AhFRO2* genes showed low expression levels in all tissues of peanut (cv. Tifrunner) under normal conditions (Figure 5). The results concurred with previous studies [25,28], suggesting that the reduction of gene expression might be beneficial for the maintenance of multiple duplicated genes and avoidance of functional redundancy.

Surviving duplicated genes would be subject to purifying selection, which could lead to divergence in both the coding and regulatory regions [30]. At the coding regions, *AhFRO2.4* from peanut and *AdFRO2.2* and *AdFRO2.3* from *A. duranensis* only have two exons, while the remaining *FRO2* genes contained eight exons. Gene/protein structures indicate that these genes appear to derive from continuous gene shortening during evolution, which may cause neofunctionalization or pseudogenization. The inducible gene expression by Fe-deficiency confirms that *AhFRO2.4* still has a function in the Fe-deficient response of peanut roots and leaves.

At the regulatory regions, CREs play essential roles in regulating gene expression through interacting with transcription factors and RNA polymerase [22]. Our results show that, although all duplicated genes of *FRO7*, *FRO8,* and some of *FRO2* (i.e., *AhFRO2.1*/*2.3*) share a similar exon–intron organization, none of them have similar CREs. The promoter of *AhFRO7.1* specifically contains TCCC-motif, LTR, and GCN4_motif, while that of *AhFRO7.2* specifically contains TCA-element, MBS, and TC-rich repeats. Similarly, the promoter of *AhFRO8.1* specifically contains AT1-motif, chs-CMA1a, chs-CMA2a, and GARE-motif, while that of *AhFRO8.2* specifically contains MRE and GCN4_motif. The differential CREs in promoters imply a divergence of transcriptional regulation between the duplicated genes.

Apart from the three short sequence proteins (AhFRO2.4, AdFRO2.2, and AdFRO2.3), all FROs contained the typical domains: Ferric_reduct, FAD_binding_8, and NAD_binding_6 (Figure 2B). Ferri_reduct domain is a ferric reductase-like transmembrane component that can transfer electrons from extracellular ferric ions to generate the reduced form of ferrous ions for transporting across the plasma membrane by specific iron transporters [12,31]. NAD- and FAD-binding domains participate in membrane electron transfer from intracellular NADPH and FAD to extracellular oxygen for superoxide production [11]. Consistent with gene structures, AhFRO2.4, AdFRO2.2, and AdFRO2.3 only contain the NAD_binding_6 domain, indicating a distinct physiological function from other homologous proteins.

AhFRO proteins were well modeled with three kinds of 3D model templates such as 6wxr.1, 8gz3.1, and 7d3f.1 (Appendix A). The best template of FRO2 for a 3D model is 7d3f.1, a cryo-EM structure of human DUOX1–DUOXA1 in a high-calcium state [32]. The best template of FRO7 is 6wxr.1, a cryo-EM structure of mouse DUOX1–DUOXA1 complex in the absence of NADPH [33]. DUOX1 is an NADPH oxidase family member that catalyzes the production of hydrogen peroxide by transferring electrons from intracellular NADPH to extracellular oxygen [32,33]. FRO8 is well modeled with 8gz3.1, the structure of human phagocyte NADPH oxidase in the resting state [34]. Phagocyte NADPH oxidase membrane-bound redox enzymes transfer electrons from intracellular NADPH to extracellular oxygen for producing superoxide anions [34]. Structural analysis indicates that AhFROs have redox activity and might reduce metal ions in different pathways.

The phylogenetic tree revealed that FRO members are grouped into three groups (I, II, and III), which is consistent with previous results [9,12]. Group I is composed of five paralogs of *AhFRO2* (*AhFRO2.1–2.5*), which exhibited considerable differences in the sequence and gene/protein structure. *AhFRO2.4* is a short sequence gene encoding 233 aa, with two TMDs, while other members contained ten TMDs. AhFRO2 is closely related to AtFRO1–3 from *Arabidopsis*. AtFRO2 is responsible for the reduction of solubilized Fe^3+^ to Fe^2+^ at the root surface in *Arabidopsis*, where Fe^2+^ is then transported into the cytoplasm via IRT1 in the root plasma membrane [10,13]. AtFRO3 localizes to mitochondrial membranes and might serve an analogous function in the mitochondrial iron homeostasis [16,17]. In this study, AhFRO2 proteins were predicted to be localized in plasma membranes, and most of *AhFRO2* genes were predominantly expressed in roots. Moreover, the expression of *AhFRO2* genes was strongly induced by Fe deficiency in both the roots and leaves of peanut seedlings. Similar results have been extensively reported in *AtFRO2* and *AtFRO3* of *Arabidopsis* [35,36]. The expression of *AhFRO2* genes in roots was significantly correlated with Fe concentrations in roots and shoots as well as the total Fe in plants, suggesting that *AhFRO2* genes might be involved in the reduction of Fe in peanut roots.

Group II contained two paralogs of *AhFRO7* (*AhFRO7.1/7.2*), which resulted from WGD events. The two paralogs are very similar in their sequence, physicochemical properties, and gene/protein structure, suggesting the same role in peanut. Phylogenetic analysis indicates that AhFRO7 is closely clustered with AtFRO6/7 from *Arabidopsis* and OsFRO1 from rice. AtFRO6 has been proven to mediate the reduction of Fe^3+^ to Fe^2+^ at the plasma membrane of leaf cells [9,14], while AtFRO7 plays a role in chloroplast iron acquisition by reducing Fe^3+^ to Fe^2+^ [15,16]. In the current study, AhFRO7 proteins were predicted to be localized in chloroplast, which is consistent with AtFRO7 in *Arabidopsis* [15,16]. Concurrent with Mukherjee et al. [9], who found that *AtFRO6* and *AtFRO7* show high expression in all the green parts of *Arabidopsis* plants, RNA-seq data showed that *AhFRO7.1*/*7.2* are highly expressed in leaves and pistils. The findings indicate a possible role for *AhFRO7.1*/*7.2* in regulating chloroplast iron acquisition. Additionally, it is thought that Fe is mainly stored in plastids of plant cells as ferritin [9]. Thus, the repression of *AhFRO7.1*/*7.2* expression in the roots under Fe deficiency might contribute to Fe translocation to leaves by reducing Fe storage in the plastids of root cells. This is illustrated by the positive correlation between the expression of *AhFRO7.1*/*7.2* and shoot Fe concentration. In contrast to roots, the expression of *AhFRO7.1*/*7.2* was induced or unaffected in the leaves. This could maintain or improve Fe reduction ability for importing into chloroplasts in leaves.

Group III included two paralogs of *AhFRO8* derived from WGD, which share the same physicochemical properties and gene/protein structure. Phylogenetic analysis indicates that AhFRO8 is closely clustered with AtFRO8 from *Arabidopsis*. Similar to AtFRO8 [16,17], AhFRO8.1/8.2 were predicted to localize to mitochondrial membranes. Unlike *AtFRO8* which is highly expressed in *Arabidopsis* shoots [9], our results show that *AhFRO8.1*/*8.2* are primarily expressed in seeds and roots (*AhFRO8.1*) of peanut. Previous studies showed that *AtFRO8* is not regulated by Fe availability [9]. However, our results show that Fe deficiency reduces the expression of *AhFRO8.1*/*8.2* in the roots of Silihong and in the leaves of both cultivars. In addition, the expression of *AhFRO8.1*/*8.2* in roots was observed to be positively correlated with shoot Fe concentrations. Although the functions of FRO8 are yet uncharacterized even in *Arabidopsis*, our data implies *AhFRO8.1*/*8.2* might be involved in mitochondrial iron homeostasis. The reduction of *AhFRO8.1*/*8.2* under Fe deficiency could reduce Fe storage in the mitochondria, leading to more Fe allocation to chloroplasts.

*FRO* genes are also assumed to be involved in copper reduction [9,10]. *Arabidopsis AtFRO2* has been shown to take a role in the reduction of Cu^2+^ to Cu^+^ at the root surface [10]. Although *AhFRO2* genes are not regulated by Cu in peanut roots, down-regulation of *AhFRO2.2/2.4* was observed in the leaves under Cu exposure. Moreover, the expression of *AhFRO2.2/2.4/2.5* positively correlated with Cu concentrations in roots and shoots as well as total Cu in plants, indicating a possible role in Cu reduction at the plasma membrane for the uptake of Cu into cells. In addition, we found that excess Cu considerably represses the expression of *AhFRO7.1/7.2* in the roots and leaves for both cultivars. The expression of *AhFRO7.1/7.2* in the roots negatively correlated with Cu concentrations in roots and shoots but positively correlated with root-to-shoot Cu translocation in peanut plants. These data suggest that *AhFRO7.1/7.2* might be involved in Cu homeostasis in peanut plants.

Interestingly, Cu and Fe could interact with each other in their accumulation and translocation in the two peanut cultivars (Table 4). Consistent with previous studies [37], we found that Fe deficiency significantly enhanced Cu concentrations in roots and shoots, and total amounts of Cu in plants, but reduced the percentage of Cu in shoots. As Fe deficiency can induce the expression of *AhFRO2.2/2.4/2.5* in roots, which positively correlated with Cu concentrations in roots and shoots as well as total Cu in plants, we assumed that *AhFRO2.2/2.4/2.5* might be responsible for higher Cu accumulation in Fe-deficient peanut plants. Similarly, the reduction of *AhFRO7.1/7.2* expression under Fe deficiency appears to decrease Cu storage in plastids of root cells and, consequently, contribute to a higher capability of Cu translocation from roots to shoots. Although Cu exposure significantly increased root Fe concentrations in Fe-sufficient peanut plants, none of the *AhFRO* genes could well explain the phenomenon.

As for the two cultivars, Silihong (Fe-deficiency tolerant cultivar) showed higher expressions of *AhFRO2.1/2.3* than Fenghua 1 (Fe-deficiency sensitive cultivar) under Fe-deficiency. Higher expressions of *AhFRO2.1/2.3* indicate a higher capacity for the reduction of Fe^3+^ to Fe^2+^ or Cu^2+^ to Cu^+^. This is in accordance with the higher concentrations of Fe and Cu in the root of Silihong under Fe deficiency compared with Fenghua 1. It is likely that higher expressions of *AhFRO2.1/2.3* contribute to Fe-deficiency tolerance in Silihong.

## 4. Materials and Methods

### 4.1. Identification of FRO Proteins in the Three Arachis Species

Protein sequences of *Arabidopsis* AtFROs (AtFRO1–8) were retrieved from a phytozome database (https://phytozome-next.jgi.doe.gov/, accessed on 2 May 2022). Using these sequences as queries, BLASTp was carried out against protein databases of *A. hypogaea* cv. Tifrunner, *A. duranensis*, and *A. ipaënsis*, which was retrieved from NCBI (https://github.com/ncbi, accessed on 10 May 2022). Non-redundant putative candidates were examined for the presence of typical conserved domains of FROs, Ferric_reduct (PF01794), FAD_binding_8 (PF08022), and NAD_binding_6 (PF08030), using the hmmscan tool (https://www.ebi.ac.uk/Tools/hmmer/search/hmmscan, accessed on 12 June 2023). Sequences containing conserved domains were used for the ClustalW alignment and phylogenetic analysis using the MEGA-X program (v. 10.2.6) together with the eight AtFROs. The phylogenetic trees were built using the neighbor-joining (NJ) method based on the Poisson model with 1000 bootstrap replicates. The proteins clustered with AtFROs were assigned as putative FRO proteins.

### 4.2. Physicochemical and Structural Characteristics of FRO Proteins

Physiochemical characteristics of FRO proteins were analyzed using the ProtParam tool (https://web.expasy.org/protparam/, accessed on 23 June 2023) [38]. The transmembrane domains (TMDs) were estimated by TOPCONS (http://topcons.net/, accessed on 18 June 2023) [39]. Subcellular targeting sites for FRO proteins were predicted using ProtComp v. 9.0 (http://www.softberry.com/berry.phtml?topic=protcomppl&group=programs&subgroup=proloc, accessed on 28 July 2023). The conserved domain of FRO proteins was detected by the Pfam tool (http://pfam.xfam.org/search#tabview=tab1, accessed on 21 July 2023) [40]. Conserved motif annotations were obtained from the MEME v. 5.3.3 (https://meme-suite.org/meme/tools/meme, accessed on 28 July 2023) [41]. Homology-modeled 3D structures of FRO proteins were predicted using the Swiss-Model (https://swissmodel.expasy.org/, accessed on 24 September 2023) [42]. 

### 4.3. Exon–intron Organization, Duplication, and Ka/Ks of FRO Genes

The exon–intron organization of *FRO* genes was identified using the GSDS (v. 2.0) (http://gsds.gao-lab.org/, accessed on 11 June 2023) [43]. One Step MCScanX integrated into the TBtools software (v. 2.034) was used for detecting the synteny relationship and duplication pattern of *FRO* genes [44]. Diagrams of exon–intron organization and gene duplication events were drawn using TBtools software [44]. *K*a/*K*s ratios were estimated by the simple *K*a/*K*s calculator (NJ) integrated into the TBtools software (v. 2.034) [44]. Based on *K*s values, the divergence time of the duplication event was calculated with the equation T = *K*s/2λ, where λ represents the neutral substitution rate that is estimated at 8.12 × 10^−9^ for peanut [18]. The CREs in promoter sequences (upstream 2.0 kb) of *AhFRO* genes were predicted by PlantCARE software (http://bioinformatics.psb.ugent.be/webtools/plantcare/html/, accessed on 1 January 2023).

### 4.4. Tissue-specific Expression Profiles of AhFRO Genes in Peanut

RNA-seq data of 22 different tissues in peanut (cv. Tifrunner) were obtained from PeanutBase (https://www.peanutbase.org/, accessed on 15 June 2022) [21]. After being transformed from read counts, TPMs (Transcripts Per Kilobase of exon model per Million mapped reads) were used as lg(TPM + 1) for constructing a heatmap diagram by Origin 2021 (v 9.8.0.200, OriginLab Corp., Northampton, MA, USA).

### 4.5. Plant Growth, Treatment, Metal Determination, and RT-qPCR Analysis

Two contrasting peanut cultivars, Fenghua 1 (Fe deficiency sensitive cultivar) and Silihong (Fe deficiency tolerant cultivar), were used for hydroponic experiments [37]. After the surface was sterilized with 5% sodium hypochlorite solution, seeds were rinsed in deionized water for 24 h at room temperature and then sown in sand for germination. Three-day-old seedlings with uniform sizes were transplanted into polyethylene pots for hydroponic culture. The culture conditions and nutrient solutions were followed as described previously by Lu et al. [45]. Ten-day-old seedlings were exposed to 0 or 10 µM CuSO_4_ under Fe-sufficient (+Fe, 50 μM Fe-EDTA) or Fe-deficient (−Fe, 0 μM Fe-EDTA) conditions, respectively. Each treatment per cultivar was repeated three times (biological replicates) with three plants per replication. Nutrient solutions were renewed twice a week during the growing period. After 14 days of treatment, plants were harvested for metal determination and RT-qPCR analysis.

The harvested roots were rinsed with 20 mM Na_2_EDTA for 15 min to remove the surface-bound metal ions and then oven-dried together with shoots. After being weighed and ground, tissue powders were digested with HNO_3_–HClO_4_ (3:1, *v*/*v*). Cu and Fe concentrations in the samples were determined by flame atomic absorbance spectrometry (WFX-110, Beijing Rayleigh Analytical Instrument Company, Beijing, China). The total Fe/Cu in plants and the percentage of Fe/Cu in shoots were calculated using the equations reported by Liu et al. [46].

Frozen tissues were used for total RNA extraction, cDNA strand synthesis, and RT-qPCR analysis, which were strictly followed according to the methods described by Tan et al. [25]. The relative mRNA abundance was normalized using the endogenous reference gene (*60S*, NCBI Entrez gene ID:112697914). The primers of *AhFROs* and *60S* are listed in Appendix A. The relative gene expression was calculated with three biological replicates using the 2^−ΔΔCT^ method [47]. Each biological replication was technically replicated three times.

### 4.6. Statistical Analysis

Data of Fe/Cu accumulation and qRT-PCR were subject to analysis of variance (ANOVA), and Duncan’s Multiple Range Test (*p* < 0.05) was used for detecting differences among group means. Pearson’s correlation analysis was used to determine the relationship between gene expression and Fe/Cu accumulation. All statistical analyses were conducted using IBM SPSS Statistics v.22 (IBM, New York, NY, USA).

## 5. Conclusions

A total of nine, four, and three *FRO* genes were identified in peanut, *A. duranensis*, and *A. ipaensis*, respectively, which were divided into three groups (I to III). Most of the *AhFRO* genes underwent WGD/segmental duplication, leading to the expansion of the *AhFRO* gene family. Clustered members generally share similar gene/protein structures. However, structural or CRE divergences and reduced expression existed in *AhFRO* genes, which may be beneficial for the maintenance of duplicate genes. *AhFRO2* and *AhFRO7* genes might be involved in the reduction of Fe/Cu in plasma membranes and chloroplast (or plastids in root cells), while *AhFRO8* genes appear to confer Fe reduction in the mitochondria. Fe deficiency-induced Cu accumulation in both cultivars, which might be associated with *AhFRO2.2/2.4/2.5* and *FRO7.1/7.2*. Our findings provide a basis for further functional characterization of *AhFRO* genes and shed new light on the possible roles of the *AhFRO* family in the Fe/Cu interaction in plants.

## Figures and Tables

**Figure 1 plants-13-00418-f001:**
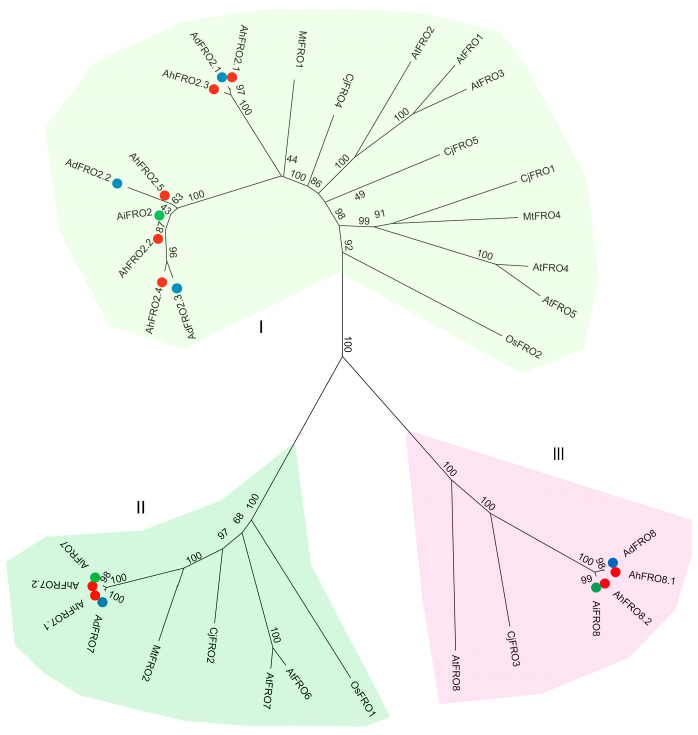
Phylogenetic relationships of FRO proteins in the three *Arachis* species and other plant species. The species involved in the evolutionary tree include *A. hypogaea* (AhFRO), *A. duranensis* (AdFRO), *A. ipaënsis* (AiFRO), *Arabidopsis thaliana* (AtFRO), *Oryza sativa* (OsFRO), *Citrus junos* (CjFRO), and *Medicago truncatula* (MtFRO). The AhFRO, AdFRO, and AiFRO proteins are marked in red, blue, and green colors, respectively.

**Figure 2 plants-13-00418-f002:**
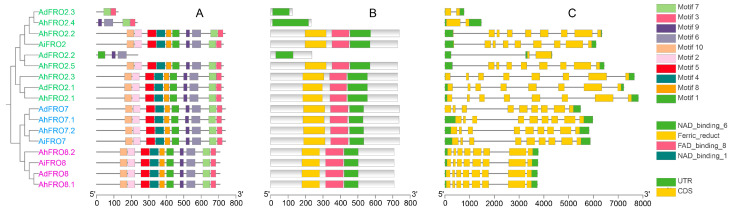
Conserved motifs (**A**) and domains (**B**) in FRO proteins and exon–intron organization of *FRO* genes (**C**) from the three *Arachis* species. UTR and CDS represent untranslated regions and coding sequences, respectively.

**Figure 3 plants-13-00418-f003:**
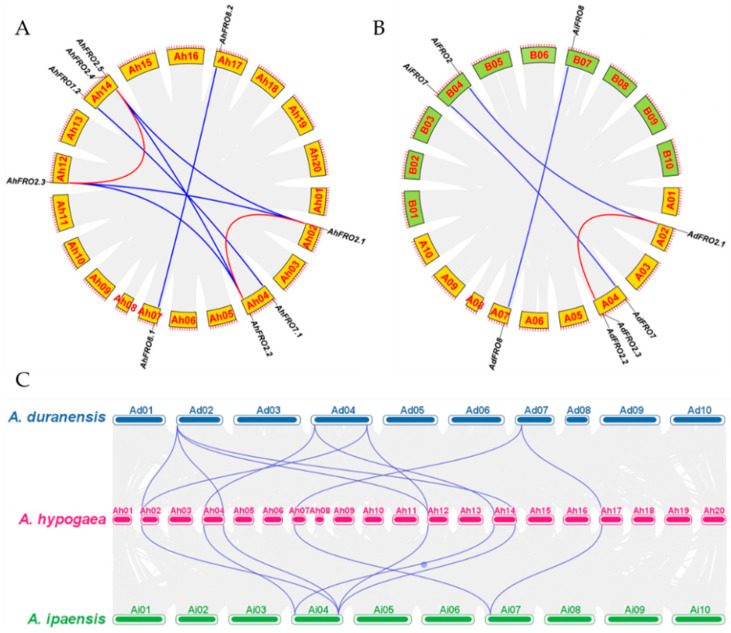
Synteny relationship of *FRO* gene pairs in the three *Arachis* species. (**A**) Synteny relationship of *AhFRO* gene pairs in *A. hypogaea*. (**B**) Synteny relationship of *FRO* gene pairs between *A. duranensis* and *A. ipaensis*. (**C**) Synteny relationship of *FRO* gene pairs among the three *Arachis* species. The red and blue lines represent segmental duplicated genes and synteny genes, respectively. The gray lines show the collinear blocks of the plant genomes.

**Figure 4 plants-13-00418-f004:**
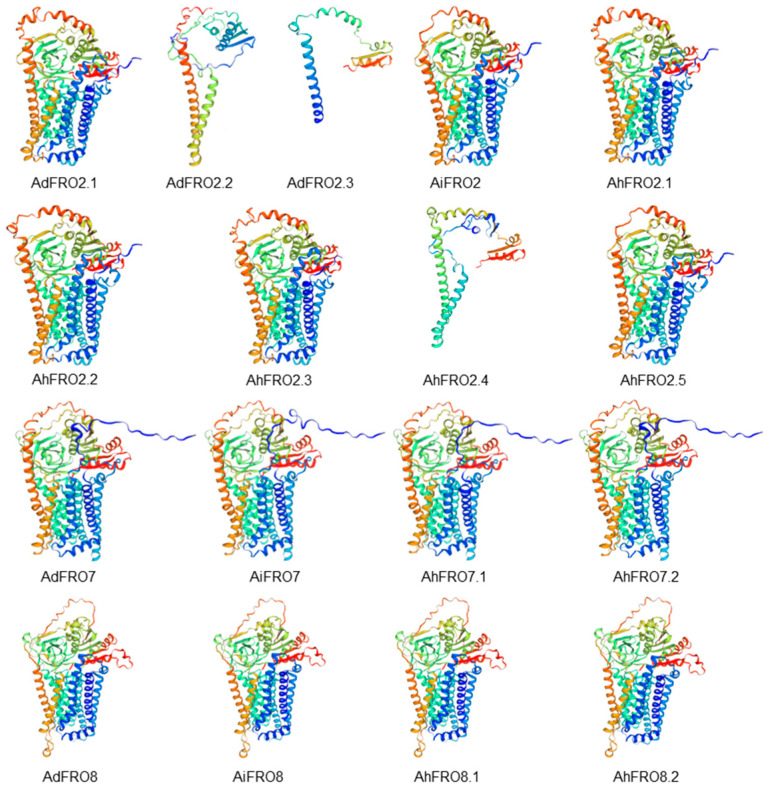
Predicted 3D structure of peanut AhFRO proteins by Swiss-Model. Models were visualized with rainbow colors from N to C terminus.

**Figure 5 plants-13-00418-f005:**
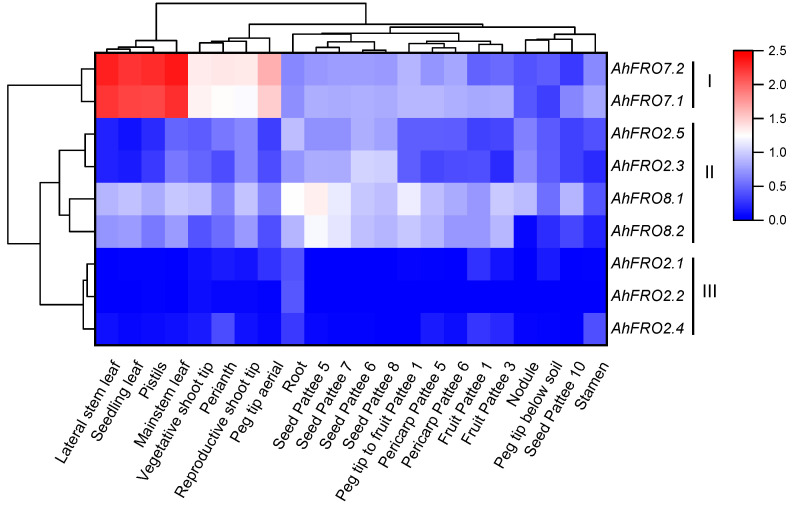
Expression profiles of *AhFRO* genes across the different tissues. Gene expression is expressed in lg(TPM + 1). Pattee 1, 3, 5, 6, 7, 8, and 10 represent differential pod developmental stages according to Pattee et al. [21].

**Figure 6 plants-13-00418-f006:**
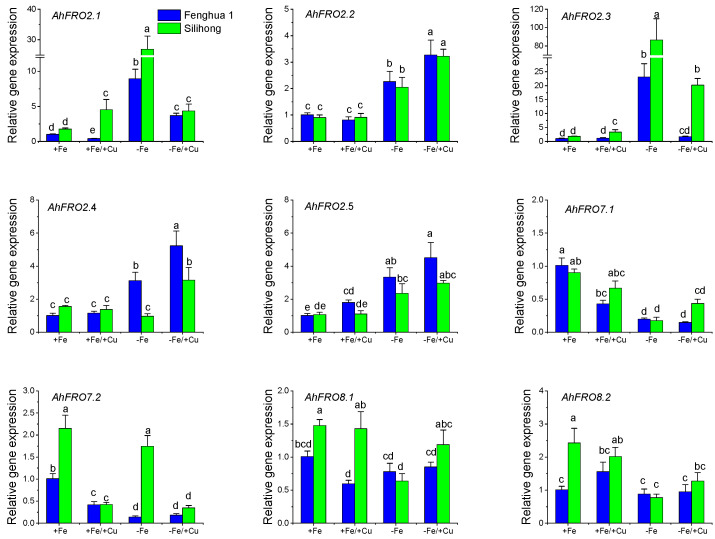
Expression levels of *AhFRO* genes in the root of two peanut cultivars in response to Fe deficiency and/or Cu exposure. Data (means ± SE, *n* = 3) sharing the same letter(s) above the error bars are not significantly different at the 0.05 level according to the Duncan multiple range test.

**Figure 7 plants-13-00418-f007:**
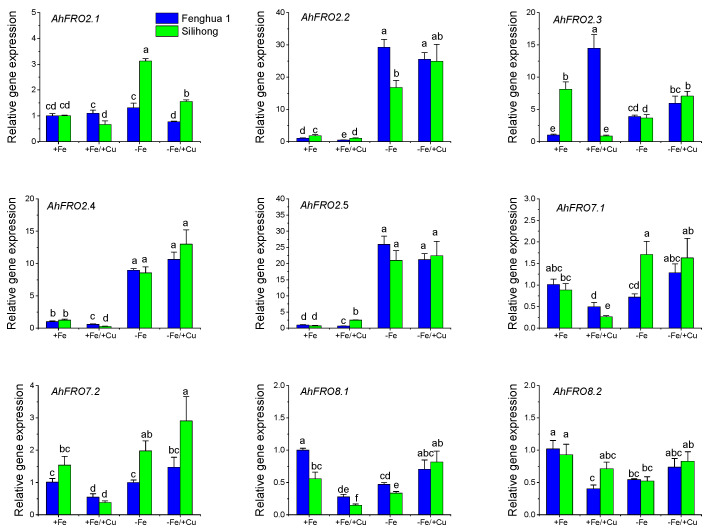
Expression levels of *AhFRO* genes in the leaves of two peanut cultivars in response to Fe deficiency and/or Cu exposure. Data (means ± SE, *n* = 3) sharing the same letter(s) above the error bars are not significantly different at the 0.05 level based on the Duncan multiple range test.

**Table 1 plants-13-00418-t001:** Molecular characterization of *FRO* genes and corresponding proteins identified in *A. hypogaea*, *A. duranensis*, and *A. ipaënsis*.

Gene Name	Gene ID	Gene Length (bp)	CDS (bp)	MW ^a^ (kDa)	aa ^b^	Instability	Aliphatic Index	GRAVY ^c^	pI ^d^	No. of TMD ^e^	Location
*AhFRO2.1*	112732410	2550	2184	81.95	727	39.95	109.13	0.368	9.38	10	PM ^f^
*AhFRO2.2*	112797510	2585	2214	83.53	737	40.69	106.85	0.187	9.47	10	PM
*AhFRO2.3*	112726301	2417	2187	82.15	728	39.81	109.52	0.348	9.25	10	PM
*AhFRO2.4*	114925155	1254	702	25.97	233	47.64	99.57	0.011	5.82	2	PM
*AhFRO2.5*	112744178	2659	2184	82.42	727	40.16	107.11	0.186	9.45	10	PM
*AhFRO7.1*	112796104	2999	2208	82.84	735	40.35	106.29	0.355	8.31	12	Chlo. ^g^
*AhFRO7.2*	112741396	2842	2217	83.11	738	41.04	105.60	0.324	8.14	12	Chlo.
*AhFRO8.1*	112702502	2590	1893	70.86	630	46.08	109.13	0.333	9.16	8	Mito. ^h^
*AhFRO8.2*	112765774	2463	2124	79.36	707	45.57	112.43	0.377	9.30	11	Mito.
*AdFRO2.1*	107472727	2407	2184	82.06	727	39.57	108.46	0.359	9.39	10	PM
*AdFRO2.2*	107485816	1056	711	26.40	236	48.31	111.48	0.244	8.23	2	PM
*AdFRO2.3*	110280250	556	372	13.52	123	35.36	91.87	−0.129	6.82	0	PM
*AdFRO7*	107483074	2508	2217	83.14	738	40.63	105.85	0.347	8.31	12	Chlo.
*AdFRO8*	107457844	2413	2124	79.45	707	44.89	111.88	0.371	9.28	10	Mito.
*AiFRO2*	107635065	2710	2184	82.49	727	40.44	107.11	0.181	9.42	10	PM
*AiFRO7*	107638776	2887	2220	83.21	739	41.14	105.98	0.329	8.14	12	Chlo.
*AiFRO8*	107609299	2446	2124	79.36	707	45.57	112.43	0.377	9.30	11	Mito.

^a^ Molecular weight, ^b^ amino acid number, ^c^ grand average of hydropathicity, ^d^ isoelectric points, ^e^ transmembrane domain, ^f^ plasma membrane, ^g^ chloroplast, ^h^ mitochondria.

**Table 2 plants-13-00418-t002:** *Ka*/*Ks* analysis of all gene duplication pairs for *AhFRO* genes.

Gene Pairs	Duplicate Type	*Ka* ^a^	*Ks* ^b^	*Ka*/*Ks* ^c^	Positive Selection	Divergence Time (Mya)
*AhFRO2.1/2.3*	Whole-genome	0.011	0.039	0.279	No	2.38
*AhFRO2.2/2.4*	Whole-genome	0.013	0.020	0.661	No	1.21
*AhFRO7.1/7.2*	Whole-genome	0.006	0.020	0.302	No	1.21
*AhFRO8.1/8.2*	Whole-genome	0.011	0.027	0.422	No	1.63
*AhFRO2.1/2.2*	Segmental	0.209	0.726	0.289	No	44.70
*AhFRO2.3/2.4*	Segmental	0.203	0.701	0.290	No	43.14

^a^ The number of nonsynonymous substitutions per nonsynonymous site, ^b^ the number of synonymous substitutions per synonymous site, ^c^ *Ka*/*Ks* ratios.

**Table 3 plants-13-00418-t003:** The cis-regulatory elements in the promoter regions of *AhFRO* genes in peanut.

Function	cis-Acting Elements	*AhFRO2.1*	*AhFRO2.2*	*AhFRO2.3*	*AhFRO2.4*	*AhFRO2.5*	*AhFRO7.1*	*AhFRO7.2*	*AhFRO8.1*	*AhFRO8.2*
Gene transcription	CAAT-box	9	20	14	19	12	8	6	13	15
	TATA-box	99	66	151	73	91	106	121	56	52
Light responsiveness	3-AF1 binding site			1						
	ACE					1				
	AE-box				1					
	AT1-motif					1	3	3	1	
	ATC-motif						1	1		
	ATCT-motif	1	1	1		1				
	Box4	5	6	5		9	2	4	3	11
	chs-CMA1a								2	
	chs-CMA2a								1	
	GA-motif	1		1						
	GATA-motif	2		3	1					
	G-box			1			3	4	5	
	GT1-motif	1	2	1		3	3	5		7
	I-box	2		2	3					
	MRE			1						1
	TCCC-motif						1		1	1
	TCT-motif	1	1	1	2	1	2	3	1	2
Phytohormone responsive	ABRE			1			3	3	3	
	CGTCA-motif			1					1	2
	GARE-motif		1			1			1	
	P-box	1	1		1					
	TCA-element					1		1	1	1
	TGACG-motif			1					1	2
	TGA-element						1			
	AuxRR-core								1	1
Abiotic stress responsive	ARE	3	2	4	3				3	8
	LTR			1			1			
	MBS	1						1		
	TC-rich repeats		1		2			2	1	1
Tissue expression	CAT-box				1		1	1		
	GCN4_motif						1			1

**Table 4 plants-13-00418-t004:** The accumulation and translocation of Fe and Cu in two peanut cultivars exposed to Fe-deficiency and/or Cu for 14 days.

Cultivars/ Treatments	[Fe]_root_ ^a^	[Fe]_shoot_ ^b^	Total Fe in Plants	% of Fe in Shoots	[Cu]_root_ ^c^	[Cu]_shoot_ ^d^	Total Cu in Plants	% of Cu in Shoots
Fenghua 1								
+Fe (control)	1203.2 ± 56.0 ce	159.5 ± 5.2 a	703.4 ± 19.2 c	46.2 ± 2.1 a	19.1 ± 2.0 e	5.5 ± 0.4 f	17.3 ± 1.5 d	65.2 ± 2.3 a
+Fe + Cu	1880.4 ± 48.8 b	109.8 ± 4.3 d	1053.4 ± 41.8 b	28.1 ± 1.4 c	168.5 ± 13.5 d	8.8 ± 0.2 d	91.2 ± 3.5 b	26.0 ± 1.9 c
−Fe	273.2 ± 8.8 f	47.1 ± 1.5 e	165.5 ± 6.9 f	47.0 ± 1.1 a	172.4 ± 6.8 d	16.9 ± 0.4 b	83.3 ± 1.5 bc	33.5 ± 2.0 b
−Fe + Cu	424.8 ± 12.4 e	48.2 ± 0.7 e	238.4 ± 21.6 e	38.1 ± 2.0 b	1742.1 ± 27.6 a	51.6 ± 1.6 a	701.2 ± 58.8 a	13.8 ± 0.6 e
Silihong								
+Fe (control)	1064.4 ± 24.9 d	121.3 ± 2.0 c	569.9 ± 15.9 d	45.1 ± 1.5 a	12.7 ± 1.2 e	3.4 ± 0.1 g	10.8 ± 0.4 d	65.7 ± 2.2 a
+Fe + Cu	2868.4 ± 43.8 a	133.2 ± 4.1 b	1240.5 ± 6.6 a	23.4 ± 0.9 d	192.8 ± 9.9 cd	6.9 ± 0.1 e	78.8 ± 1.8 bc	19.1 ± 0.8 d
−Fe	504.2 ± 48.2 e	44.2 ± 2.5 e	167.1 ± 5.9 f	34.7 ± 1.0 b	199.9 ± 3.3 c	13.4 ± 0.6 c	61.7 ± 5.8 c	28.8 ± 0.7 c
−Fe + Cu	473.9 ± 31.9 e	50.2 ± 3.0 e	275.2 ± 25.8 e	35.7 ± 1.3 b	1649.3 ± 24.8 b	54.4 ± 2.4 a	719.3 ± 19.3 a	14.7 ± 0.4 de
ANOVA (F value)								
Cu	582.2 ***	11.1 **	397.4 ***	131.3 ***	6515.8 ***	752.6 ***	516.9 ***	729.6 ***
Fe	2451.4 ***	1326.4 ***	2038.3 ***	9.3 **	6602.5 ***	1381.0 ***	481.0 ***	369.4 ***
Cultivar (Cv)	109.6 ***	3.0 ns	2.3 ns	23.7 ***	1.3 ns	2.6 ns	0.1 ns	5.4 *
Cu × Fe	478.7 ***	23.9 ***	194.0 ***	58.5 ***	4204.2 ***	525.8 ***	330.7 ***	137.8 ***
Cu × Cv	76.7 ***	52.3 ***	34.9 ***	2.3 ns	4.7 ∗	4.8 *	0.3 ns	0.1 ns
Fe × Cv	27.8 ***	2.2 ns	0.1 ns	4.5 *	4.0 ns	1.2 ns	0.1 ns	0.3 ns
Cu × Fe × Cv	147.2 ***	38.0 ***	22.4 ***	10.4 **	13.2 **	3.9 ns	0.5 ns	8.6 *

^a^ Fe concentration in roots, ^b^ Fe concentration in shoots, ^c^ Cu concentration in roots, ^d^ Cu concentration in shoots. Data (means ± SE, *n* = 3) sharing the same letter(s) in the same column are not significantly different at the 0.05 level based on the Duncan multiple range test. * *p* < 0.05, ** *p* < 0.01, *** *p* < 0.001, ns, not significant.

**Table 5 plants-13-00418-t005:** Pearson’s correlation analysis (*r* value, *n* = 24) of metal accumulation and the expression of *AhFRO* genes in the roots and leaves of Fenghua 1 and Silihong.

Gene Expression	[Fe]_root_ ^a^	[Fe]_shoot_ ^b^	Total Fe in Plants	% of Fe in Shoots	[Cu]_root_ ^c^	[Cu]_shoot_ ^d^	Total Cu in Plants	% of Cu in Shoots
Roots								
*AhFRO2.1*	−0.345	−0.526 **	−0.476 *	−0.059	−0.113	−0.034	−0.144	−0.192
*AhFRO2.2*	−0.678 **	−0.773 **	−0.714 **	0.173	0.794 **	0.843 **	0.772 **	−0.550 **
*AhFRO2.3*	−0.380	−0.519 **	−0.481 *	−0.038	−0.084	0.003	−0.103	−0.161
*AhFRO2.4*	−0.488 *	−0.567 **	−0.500 *	0.199	0.748 **	0.763 **	0.711 **	−0.450 *
*AhFRO2.5*	−0.603 **	−0.755 **	−0.620 **	0.164	0.660 **	0.702 **	0.624 **	−0.546 **
*AhFRO7.1*	0.451 *	0.856 **	0.521 **	0.146	−0.437 *	−0.497 *	−0.415 *	0.713 **
*AhFRO7.2*	−0.071	0.222	−0.079	0.255	−0.461 *	−0.477 *	−0.471 *	0.649 **
*AhFRO8.1*	0.301	0.345	0.269	−0.013	−0.013	−0.058	−0.020	0.252
*AhFRO8.2*	0.511 *	0.452 *	0.515 **	−0.203	−0.247	−0.312	−0.253	0.273
Leaves								
*AhFRO2.1*	−0.405 *	−0.495 *	−0.488 *	−0.050	−0.072	0.008	−0.087	−0.108
*AhFRO2.2*	−0.754 **	−0.865 **	−0.800 **	0.256	0.618 **	0.707 **	0.622 **	−0.498 *
*AhFRO2.3*	−0.017	−0.129	0.114	−0.212	0.115	0.103	0.140	−0.113
*AhFRO2.4*	−0.762 **	−0.868 **	−0.809 **	0.184	0.735 **	0.812 **	0.742 **	−0.530 **
*AhFRO2.5*	−0.734 **	−0.889 **	−0.806 **	0.193	0.548 **	0.646 **	0.547 **	−0.518 **
*AhFRO7.1*	−0.639 **	−0.492 *	−0.647 **	0.257	0.440 *	0.473 *	0.440 *	−0.107
*AhFRO7.2*	−0.585 **	−0.486 *	−0.591 **	0.238	0.506 *	0.536 **	0.525 **	−0.130
*AhFRO8.1*	−0.474 *	0.053	−0.372	0.595 **	0.350	0.354	0.365	0.325
*AhFRO8.2*	−0.031	0.357	0.000	0.386	0.086	0.031	0.100	0.452 *

^a^ Fe concentration in roots, ^b^ Fe concentration in shoots, ^c^ Cu concentration in roots, ^d^ Cu concentration in shoots, * *p* < 0.05, ** *p* < 0.01.

## Data Availability

Data are contained within the article.

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
