# Peer review of "Genome-Wide Identification of the Ferric Chelate Reductase (FRO) Gene Family in Peanut and Its Diploid Progenitors: Structure, Evolution, and Expression Profiles"

_plants, 2024, doi:10.3390/plants13030418_

Round 1
Reviewer 1 Report
Comments and Suggestions for Authors
The manuscript “Genome-wide identification of the ferric chelate reductase (FRO) gene family in peanut and its diploid progenitors: structure, evolution, and expression profiles” deals with the characterization of FRO genes in A. hypogaea and two diploid progenitors.
Some revisions are reported below:
Abstract
The Abstract is unclear in presenting the main findings and conclusions. The authors have listed the genes identified and their expression, I believe that the list of genes should be avoided. It’s necessary to better explain the method used, results and their potentialities.
In line 23-24 you introduced the expressions genes in Silihong. This sentence is confusing, because at the beginning you discussed about the study of A. hypogea, A. duranensis and A. ipaensis.
Rephrase the conclusion (lines 24-25), it is not obvious.
Results
The figure 2 isn’t clear. You should use non-shadow colors.
The Table 3 should be formatted.
In the caption of the figure 6 you should replace “roots” to “root” (line 245).
In the results (paragraph 2.7) you should introduce why you used for the expression analysis Fenghua 1 and Silihong.
Discussion
The discussion about evolution and structure of FRO genes is too long.
In the discussion lack the description of results about tissues expression of FRO genes.
Material and Methods
Lines 524. The reference gene utilized is not clear. What the number 112697914 indicates?
Conclusions
You rewrite exactly the sentences reported in the abstract and results section. You should rewrite the conclusions emphasizing the importance of results obtained.
When you indicate a protein name the suffix Ah, At, etc. should be in italics.
Author Response
Response to Review 1
Abstract
The Abstract is unclear in presenting the main findings and conclusions. The authors have listed the genes identified and their expression, I believe that the list of genes should be avoided. It’s necessary to better explain the method used, results and their potentialities.
Response: Thanks for your comments. According to your suggestion, we have revised the Abstract (lines 10-15, 22-25).
In line 23-24 you introduced the expressions genes in Silihong. This sentence is confusing, because at the beginning you discussed about the study of A. hypogea, A. duranensis and A. ipaensis.
Response: I'm sorry for this oversight. We have added the method in the Abstract (lines 13-15).
Rephrase the conclusion (lines 24-25), it is not obvious.
Response: Thanks for your comments. The sentence has been revised (lines 24-25).
Results
The figure 2 isn’t clear. You should use non-shadow colors.
Response: Thanks for your comments. The figure 2 has been re-plotted according to your suggestion (Figure 2).
The Table 3 should be formatted.
Response: Thanks for your comments. Table 3 has been formatted according to your suggestion (Table 3).
In the caption of the figure 6 you should replace “roots” to “root” (line 245).
Response: Replaced (line 243).
In the results (paragraph 2.7) you should introduce why you used for the expression analysis Fenghua 1 and Silihong.
Response: Revised (line 232-234).
Discussion
The discussion about evolution and structure of FRO genes is too long.
Response: Thanks for your comments. We completely agree with your comments. According to your suggestion, we have shortened discussion about evolution and structure of FRO genes in the revised manuscript (lines 296-297, 306-317, 327, 340).
In the discussion lack the description of results about tissues expression of FRO genes.
Response: Thanks for your comments. The tissue-specific expression of FRO genes has been discussed (lines 372-373, 388-389, 401-403).
Material and Methods
Lines 524. The reference gene utilized is not clear. What the number 112697914 indicates?
Response: Thanks for your comments. According to your suggestion, the number of the reference gene have been added and some descriptions been detailed in the revised version of manuscript (lines 507-512).
Conclusions
You rewrite exactly the sentences reported in the abstract and results section. You should rewrite the conclusions emphasizing the importance of results obtained.
Response: Thanks for your comments. According to your suggestion, we have rewritten the conclusions in the revised version of manuscript (lines 522-531).
When you indicate a protein name the suffix Ah, At, etc. should be in italics.
Response: Thanks for your comments. According to your suggestion, the prefix (Ah, At, etc.) of all protein names have been changed to italics and highlighted in red.

Reviewer 2 Report
Comments and Suggestions for Authors
This manuscript is about an interesting study presenting a genome-wide identification of the ferric chelate reductase gene family in peanut, including data regarding structure and evolution, as well as expression profiles under 4 different Fe/Cu conditions. This work is very well designed and the results are clearly presented and of high interest to plant scientists working with iron homeostasis in plants. Nevertheless, there are a number of shortcomings in primer design that need to be addressed, before evaluating gene expression data.
More specifically, when I used NCBI primer blast to check the various primer pairs, I noticed the following:
- Primer pair designed for AhFRO2.2 seems to give two products with different lenghts (164 and 134 bp).
- Primer pair designed for AhFRO2.3 seems to give two products with different lenghts (211 and 208 bp). Moreover, the smaller one is on a target template that the primer pair designed for AhFRO2.1 also gives a product (XM_025781129.2).
- Primer pair designed for AhFRO2.4 seems to give products on the same target templates as the respective primer pair of AhFRO2.2 (XM_029292324.1, XM_029292325.1, XM_029292326.1, XM_025793726.2, XM_025840491.2, XM_025840492.2, XM_025840493.2).
- Primer pair designed for AhFRO2.5 seems to give two products with different lenghts (163 and 133 bp). Moreover, this primer pair seems to give products on the same target templates as the respective primer pair of AhFRO2.2 and AhFRO2.4 (XM_029292324.1, XM_029292325.1, XM_029292326.1, XM_025793726.2, XM_025840491.2, XM_025840492.2, XM_025840493.2).
- Primer pairs designed for AhFRO8.1 and AhFRO8.2 also give products on the same target templates (XM_025753557.2, XM_025753558.2, XM_025753559.2, XM_025811631.2, XM_025811632.2).
The above mentioned issues with the various primer pairs have to be addressed before evaluating gene expression data.
Round 2
Reviewer 2 Report
Comments and Suggestions for Authors
I would like to thank the authors for their responses to my concerns. I understand that it is a rather difficult task to design specific primer pairs for genes that have so similar DNA sequences. However, I believe that in some cases, such as that of AhFRO2.3, a single different base in one of the two primers does not necessarily mean that the formation of the second product (AhFRO2.1) is also prevented. On the other hand, data presented in Figures 6 and 7 further supports what you claim. Thank you for this beautiful work.